# Saliva as a testing specimen with or without pooling for SARS-CoV-2 detection by multiplex RT-PCR test

Qing Sun[1]*, Jonathan Li[1], Hui Ren[1], Larry Pastor[1], Yulia Loginova[1], Roberta Madej[1], Kristopher Taylor[2], Joseph K. Wong[2], Zhao Zhang[1], Aiguo Zhang[1], Chuanyi M. Lu[2]*, Michael Y. Sha[1]*

**1** DiaCarta Inc., Richmond, California, United States of America, **2** University of California and VA Healthcare System, San Francisco, California, United States of America

* qsun@diacarta.com (QS); mark.lu@va.gov (CML); msha@diacarta.com (MYS)

## Abstract

**Data Availability Statement:** All relevant data are within the manuscript and its Supporting information files.

### Background

Sensitive and high throughput molecular detection assays are essential during the coronavirus disease 2019 (COVID-19) pandemic, caused by the severe acute respiratory syndrome coronavirus 2 (SARS-CoV-2). The vast majority of the SARS-CoV-2 molecular assays use nasopharyngeal swab (NPS) or oropharyngeal swab (OPS) specimens collected from suspected individuals. However, using NPS or OPS as specimens has apparent drawbacks, e.g. the collection procedures for NPS or OPS specimens can be uncomfortable to some people and may cause sneezing and coughing which in turn generate droplets and/or aerosol particles that are of risk to healthcare workers, requiring heavy use of personal protective equipment. There have been recent studies indicating that self-collected saliva specimens can be used for molecular detection of SARS-CoV-2 and provides more comfort and ease of use for the patients. Here we report the performance of QuantiVirus™ SARS-CoV-2 test using saliva as the testing specimens with or without pooling.

### Methods

Development and validation studies were conducted following FDA-EUA and molecular assay validation guidelines. Using SeraCare Accuplex SARS-CoV-2 reference panel, the limit of detection (LOD) and clinical performance studies were performed with the QuantiVirus™ SARS-CoV-2 test. For clinical evaluation, 85 known positive and 90 known negative clinical NPS samples were tested. Additionally, twenty paired NPS and saliva samples collected from recovering COVID-19 patients were tested and the results were further compared to that of the Abbott m2000 SARS-CoV-2 PCR assay. Results of community collected 389 saliva samples for COVID-19 screening by QuantiVirus™ SARS-CoV-2 test were also obtained and analyzed. Additionally, testing of pooled saliva samples was evaluated.

**Funding:** The authors received no specific funding for this work. This study was conducted by DiaCarta R&D and does not involve extramural funding. JW, KT and CML provided leftover and deidentified clinical specimens including testing validation samples and helped data analysis and manuscript preparation. DiaCarta didn't provided financial compensation to JW, KT and CML. The funder Diacarta Inc provided support in the form of salaries for authors [QS, HR, JL, LP, RM, YL, ZZ, AZ and MS], but did not have any additional role in the study design, data collection and analysis, decision to publish, or preparation of the manuscript. The specific roles of these authors are articulated in the 'author contributions' section.

**Competing interests:** This commercial affiliation (to Diacarta Inc) does not alter our adherence to PLOS ONE policies on sharing data and materials.

## Results

The LOD for the QuantiVirus™ SARS-CoV-2 test was confirmed to be 100–200 copies/mL. The clinical performance studies using contrived saliva samples indicated that the positive percentage agreement (PPA) of the QuantiVirus™ SARS-CoV-2 test is 100% at 1xLOD, 1.5xLOD and 2.5xLOD. No cross-reactivity was observed for the QuantiVirus™ SARS-CoV-2 test with common respiratory pathogens. Testing of clinical samples showed a positive percentage agreement (PPA) of 100% (95% CI: 94.6% to 100%) and a negative percentage agreement (NPA) of 98.9% (95% CI: 93.1% to 99.9%). QuantiVirus™ SARS CoV-2 test had 80% concordance rate and no significant difference ($p = 0.13$) between paired saliva and NPS specimens by Wilcoxon matched pairs signed rank test. Positive test rate was 1.79% for 389 saliva specimens collected from local communities for COVID-19 screening. Preliminary data showed that saliva sample pooling up to 6 samples (1:6 pooling) for SARS-CoV-2 detection is feasible (sensitivity 94.8% and specificity 100%).

## Conclusion

The studies demonstrated that the QuantiVirus™ SARS-CoV-2 test has a LOD of 200 copies/mL in contrived saliva samples. The clinical performance of saliva-based testing is comparable to that of NPS-based testing. Pooling of saliva specimens for SARS-CoV-2 detection is feasible. Saliva based and high-throughput QuantiVirus™ SARS-CoV-2 test offers a highly desirable testing platform during the ongoing COVID-19 pandemic.

## Introduction

A novel coronavirus, severe acute respiratory syndrome coronavirus 2 (SARS-CoV-2, previously provisionally named 2019 novel coronavirus or 2019-nCoV), has been identified as the cause of respiratory infection including severe pneumonia outbreak that started in Wuhan, China in late 2019 [1, 2], and has since become a global pandemic. The disease was named the coronavirus disease of 2019 (COVID-19) by the World Health Organization in February 2020. It has been determined that SARS-CoV-2 can be transmitted from person-to-person (symptomatic or asymptomatic) and is more transmissible than SARS-CoV [3–5].

Nasopharyngeal swab (NPS) and oropharyngeal swab (OPS) samples are widely accepted as specimens for the detection of SARS–CoV–2 since the start of the COVID-19 pandemic. However, the collection procedures for NPS and OPS specimens may cause discomfort and, in some people, sneezing and coughing. The latter in turn can generate droplets or aerosol particles that place healthcare workers collecting these specimens at risk [6], requiring heavy use of personal protective equipment (PPE). Poor tolerability of NPS and OPS sampling can result in false-negative tests due to inadequate or poor quality of specimen collection [7–10]. Recent investigations by Wyllie et al. [11] and Hanson et al. [12] suggested that saliva is a viable and even more sensitive alternative to NPS specimens, and could also enable at-home self-administered sample collection for large-scale SARS-CoV-2 molecular testing. Other researchers [13] also reported that SARS-CoV-2 was detected in 91.7% (n = 11) of the initial saliva specimens from confirmed COVID-19 patients. All saliva specimens (n = 33) collected from patients whose NPS specimens tested negative for COVID-19 also tested negative. It is apparent that detection of SARS CoV-2 in saliva can be used as a more appealing and cost-effective

alternative for the diagnosis of COVID-19. Indeed, a molecular test using saliva samples was first approved for FDA under EUA on May 8, 2020 [14].

The use of saliva specimens might decrease the risk of nosocomial transmission of COVID-19 and is ideal for situations in which NPS or OPS specimen collection may be impractical [15–18]. Collecting saliva is easy and more tolerable to patients, can reduce risk of cross-infection, and can be used in settings where PPE is not readily available. It will also be useful for testing infants and young children in daycare facilities and schools.

The QuantiVirus™ SARS-CoV-2 Test is a real-time reverse transcription polymerase chain reaction (RT-qPCR) test that includes the assay controls for the qualitative detection of viral RNA from SARS-CoV-2 in NPS, OPS, saliva or sputum specimens collected from patients who are suspected of COVID-19 infection. Extracted RNA is reverse-transcribed and amplified in a single reaction. In this multiplex qPCR method, the Orf1ab, N, and E genes of the SARS-CoV-2 genome are targeted in the RT-PCR assay (Fig 1A). Primers and TaqMan probes designed for conserved regions of the SARS-CoV-2 virus genome allow specific amplification and detection of the viral RNA from all variants of SARS-CoV-2 from respiratory specimens. The Human RNase P gene is used as an Internal Control (IC) to monitor viral RNA extraction

(A)

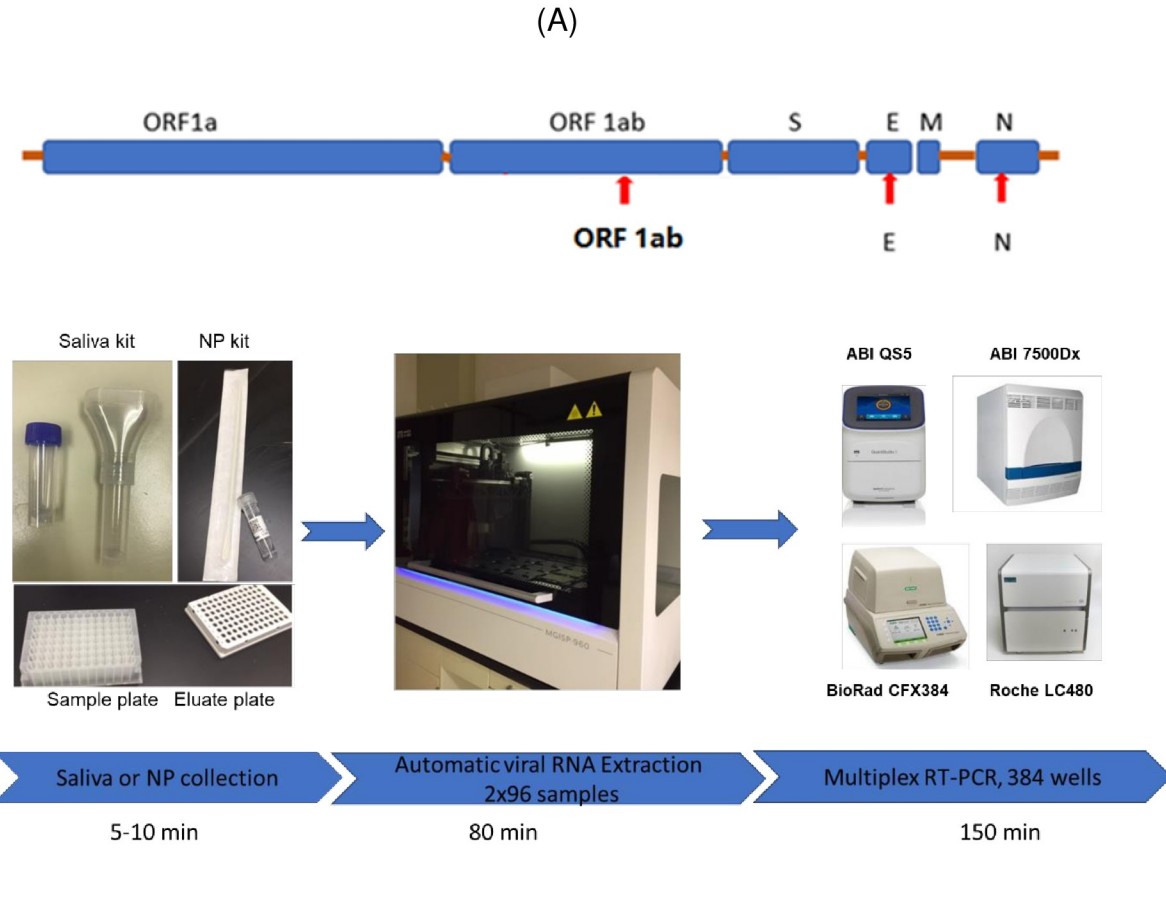

(B)

**Fig 1.** (A) SARS-CoV-2 genome structure and assay target genes. Republished from FDA EUA [14] under a CC BY license, with permission from Diacarta Inc, original copyright [2020]. (B) a high throughput workflow for SARS-COV-2 detection from sample collection to result availability within about 4 hrs.

efficiency and assess amplifiable RNA in the samples to be tested. The test is a multiplex RT-PCR assay consisting of one reaction with primers and probes for the viral gene targets (Orf1ab, N and E genes) and IC in one tube, designed to increase assay throughput.

We demonstrate here that saliva sampling is an adequate alternative to NPS and OPS sampling and can be used for COVID-19 testing using the QuantiVirus SARS-CoV-2 test.

## Methods

### Study design and ethics

Besides contrived saliva samples, deidentified leftover patient NPS and saliva samples were used in the study. All patient specimens were collected in May-September 2020 and previously tested at UCSF affiliated San Francisco VAMC clinical laboratories and DiaCarta's CLIA laboratory for clinical diagnostic or screening purpose. Other than qualitative RT-PCR results (positive or negative), only PCR cycle threshold (Ct) values were included in study analysis and no patient clinical chart reviews were performed. This study was approved by the institutional review board (IRB) at UCSF (UCSF IRB #11–05207) as a no-subject contact study with waiver of consent and as exempt under category 4.

### Clinical specimens

Clinical samples were collected from patients who had previously been tested positive for SARS-CoV-2. Paired NPS and saliva samples were collected at the same time. The QuantiVirus™ Saliva Collection Kit (DiaCarta, Inc. cat# DC-11-0021) was used for saliva collection, following the kit insert instructions and under the supervision of healthcare providers. No eat or drink 30 minutes before saliva sample collection.

Each saliva sample contains about 2 mL liquid saliva and 2 mL viral transport media. The NPS and saliva samples are refrigerated and processed for testing within 24 hours after collection.

### Sample pooling

Positive saliva and negative saliva samples were pooled together according to the experiment design for 1:6 (i.e., 1 positive mixed with 5 negatives) and 1:12 (i.e., 1 positive mixed with 11 negatives) pooling, respectively. A total of 77 positive patient samples and 385 negative samples were used for pooling at 1:6 ratio to create 77 pooled positive samples and 54 pooled negative samples. After mixing the pooled samples, RNA was extracted for RT-PCR according to the testing protocol.

### Viral RNA extraction

MGI's automatic RNA/DNA extraction instrument MGISP-960 (MGI Tech Co., Ltd, China) was used for the SARS-CoV-2 viral RNA extraction according to the manufacturer's instructions, for which 200 μL of each NPS VTM or saliva sample was used. For each batch of clinical samples to be tested, an extraction control (EC) was included (spike 20 μL of EC from the QuantiVirus™ SARS-CoV-2 kit into 180 μL sterile RNase-free water). The clinical samples and spiked EC were processed and extracted on the MGI platform. The extraction output is RNA in 30–50 μL RNase-free water, 5.5 μL of which is used for the PCR reaction per test. The turnaround time from sample extraction to PCR final report is around 4 hrs (Fig 1B). Precautions were taken while handling extracted RNA samples to avoid RNA degradation. Extracted RNA samples were stored at -80˚C if not immediately used for RT-PCR.

## Multiplex primer and probe design

Target gene sequences in the SARS-CoV-2 genome, the N gene, E gene and ORF1ab gene were identified and selected for test development. The gene sequences were retrieved from GenBank and GISAID databases for primer and probe designs to ensure coverage of all SARS-CoV-2 strains. Multiple alignments of the collected sequences were performed using Qiagen CLC Main Workbench 20.0.4., and conserved regions in each target gene were identified using BioEditor 7.2.5. prior to primer and probe designs. Primers and probes were designed to target the most conserved regions of each of the target genes of the viral genome, using Primer3plus software and following general rules of real-time PCR design. All primers were designed with a melting temperature (Tm) of approximately 60˚C and the probes were designed with a Tm of about 65˚C. The amplicon sizes were kept as short as possible within the range of 70 bp to 150 bp for each primer pair to achieve better amplification efficiency and detection sensitivity. All primers and probes were synthesized by Integrated DNA Technologies, Inc. (IDT, Coralville, IA, USA) and LGC Biosearch Technologies (Novato, CA, USA), respectively.

## Real-time reverse-transcription PCR (rRT-PCR)

The total volume of one RT-PCR reaction for all targets is 10 μL, including 5.5 μL of RNA, 2.0 μL of 5x primer and probe mixture (final concentration of 0.2 μM and 0.1 μM, respectively), and 2.5 μL of 4x TaqPath™ 1-Step RT-qPCR Master Mix (Catalog number A28526, Thermo Fisher, Waltham, MA) or 4x Inhibitor-Tolerant RT-qPCR mix (MDX016-50, Meridian Bioscience, Tennessee). Thermal cycling was performed at 25˚C for 2 min for uracil-N-glycosylase gene (UNG) incubation and 53˚C for 10 min for reverse transcription, followed by 95˚C for 2 min and then 45 cycles of 95˚C for 3 sec, and 60˚C for 30 sec. QuantStudio™ 5 Real-Time PCR System (Thermo Fisher, USA), Applied Biosystems™ 7500 Fast Dx Real-Time PCR Instrument (Thermo Fisher, USA), BioRad CFX384 (Bio-Rad, USA) and Roche LightCycler 480 II (Roche, USA) were used for rRT-PCR amplification and detection.

## Analytical sensitivity and limit of detection (LOD)

To determine the Limit of Detection (LoD) and analytical sensitivity of the QuantiVirus SARS CoV-2 Test kit, studies using empirical method were performed using serial dilutions of analyte and the LoD was determined to be the lowest concentration of template that could reliably be detected with 95% of all tested positive. LoD of each target assay in the QuantiVirus™ SARS-CoV-2 Test were conducted and verified using SeraCare AccuPlex SARS-CoV-2 Reference Material Kit (Cat# 0505–0126). Non- infectious viral particles from the AccuPlex SARS-CoV-2 Reference Material Kit were spiked in saliva at various concentrations (50 copies/mL, 100 copies/mL, and 200 copies/mL) diluted from the stock concentration of 5000 copies/mL. Real- time RT-PCR assay was performed with the provided kit reagents and tested triplicate on ABI QS5, ABI 7500 Fast Dx, Bio-Rad CFX 384 PCR and Roche LightCycler 480 II instruments. Then the LOD was confirmed by testing 1xLoD of viral RNA with 20 replicates. The LoD was determined to be the lowest concentration (copies/ml) at which ≥95% (19/20) of the 20 replicates were tested as positive.

## Precision

Precision studies include intra-run, inter-run, instrument, and operator variability evaluation. The assay precision was assessed by the repeat testing of samples with three or more different template concentrations. (1) Inter-assay %CV was established for same lot of reagents tested

on the same instrument by the same user; (2) Intra-assay %CV was established through performance of kit on reference samples run in replicates of nine; and (3) Operator variability was evaluated with one lot of reagents by two operators. Reproducibility is demonstrated based on %CV of Ct values.

### Microorganism panel for cross-reactivity

MERS- coronavirus, SARS-CoV coronavirus samples were ordered from IDT. NATtrol Respiratory Validation Panel was ordered from ZeptoMetrix (cat# NATRVP-3, Buffalo, NY). RNA/DNA were extracted from high titer stocks of the potentially cross-reacting microorganisms.

### Statistical data analysis

Average cycle threshold (Ct), standard deviation (SD) and coefficient of variation (CV) were calculated using Microsoft Office Excel 365 software (Microsoft, Redmond, WA). Clinical sensitivity, specificity, positive predictive value (PPV) and negative predictive value (NPV) at two-sided 95% confidence interval (CI) were analyzed using MedCalc software Version 19.3.1. NP and saliva pair analysis was conducted by Wilcoxon signed rank test.

## Results

### Validation of QuantiVirus™ SARS-CoV-2 test kit

**Analytical sensitivity.** Non-infectious viral particles from the AccuPlex SARS-CoV-2 Reference Material Kit (SeraCare Bioscience) were spiked in saliva at various concentrations (50, 100 and 200 copies/mL). Real-time RT-PCR assay was performed with the provided kit reagents. The assessment of individual assay result is that sample Ct <40 indicates positive and Ct>40 indicates negative. Therefore, 100 copies/mL were determined as a tentative LOD due to 50 copies/mL sample was undetectable from E gene target (S1 Table).

We then validated the QuantiVirus™ SARS-CoV-2 kit on four qPCR instruments from different vendors, using contrived saliva samples by 20 measurements. The overall analytical sensitivity (lower limit of detection or LOD) is around 100–200 copies/mL under 95% confidence interval (Table 1).

The validation data established that the LOD of the assay is 200 copies/mL on ABI 7500 Fast Dx (Table 1a), 100 copies/mL on Bio-Rad CFX 384 (Table 1b), 200 copies/mL on Roche LightCycler 480 II (Table 1c), and 200 copies/mL on the ThermoFisher ABI QuantStudio 5 (Table 1d).

**Assay precision.** *Intra-Assay*. QuantiVirus SARS CoV-2 assay at four different sample template concentrations (100, 200, 300 and 500 copies/mL) was repeated 10 times and run on the sample 384-well plate on BioRadCFX instrument. Average Ct and CV were calculated and summarized in S2a Table. The Intra assay overall CV was <3% and acceptable for this assay.

*Operator reproducibility*. The QuantiVirus SARS CoV-2 Test reactions were set up by two operators using the same lot of reagent and run on the ABI QuantStudio 5 instrument. Average Ct and CV were calculated and summarized in S2b Table. Overall CV for two operators is <3% and is acceptable for this assay.

*Inter-instrument reproducibility*. The QuantiVirus SARS CoV-2 Assay reactions were set up with three replicates and run on three different qPCR instruments including Bio-Rad CFX 384, ABI QS5 and ABI 7500 Fast Dx. Average Ct and CV were calculated and summarized in S2c Table. The results indicate that three instruments have <5% CV and is acceptable.

**Cross-reactivity (assay specificity).** We tested the cross-reactivity as part of the assay development. MERS-coronavirus and SARS-CoV coronavirus samples were ordered from

**Table 1.** a. Results of twenty replicates for analytical sensitivity confirmation on the ABI 7500 Dx. b. Results of twenty replicates for analytical sensitivity confirmation on the BioRad CFX 384. c. Results of twenty replicates for analytical sensitivity confirmation on the Roche LC 480. d. Results of twenty replicates for analytical sensitivity confirmation on the ABI QS5.

| Target | RNA (copy/mL) | No. Replicates (N) | Avg Ct | SD | CV | Positive | Negative | Call Rate |
|---|---|---|---|---|---|---|---|---|
| **a** | | | | | | | | |
| ORF1ab gene | 100 copies/mL | 20 | 34.28 | 1.05 | 3.08% | 20 | 0 | 100% |
| N gene | 100 copies/mL | 20 | 35.73 | 1.12 | 3.13% | 20 | 0 | 100% |
| E gene | 200 copies/mL | 20 | 34.24 | 0.98 | 2.87% | 20 | 0 | 100% |
| **b** | | | | | | | | |
| ORF1ab gene | 100 copies/mL | 20 | 33.76 | 0.97 | 2.87% | 20 | 0 | 100% |
| N gene | 100 copies/mL | 20 | 35.97 | 1.02 | 2.85% | 20 | 0 | 100% |
| E gene | 100 copies/mL | 20 | 37.87 | 0.58 | 1.52% | 20 | 0 | 100% |
| **c** | | | | | | | | |
| ORF1ab gene | 100 copies/mL | 20 | 32.85 | 0.57 | 1.7% | 20 | 0 | 100% |
| N gene | 200 copies/mL | 20 | 35.04 | 0.58 | 1.7% | 20 | 0 | 100% |
| E gene | 100 copies/mL | 20 | 36.13 | 0.59 | 1.6% | 20 | 0 | 100% |
| **d** | | | | | | | | |
| ORF1ab gene | 200 copies/mL | 20 | 34.09 | 0.66 | 1.92% | 20 | 0 | 100% |
| N gene | 200 copies/mL | 20 | 35.11 | 1.81 | 5.14% | 20 | 0 | 100% |
| E gene | 200 copies/mL | 20 | 34.99 | 1.68 | 4.82% | 20 | 0 | 100% |

Table 1a-1d were republished from FDA EUA [14] under a CC BY license, with permission from Diacarta Inc, original copyright [2020].

IDT and NATtrol Respiratory Verification Panel from ZeptoMetrix (cat#NATRVP-3). RNA/DNA were extracted from high titer stocks of the potentially cross-reacting microorganisms (estimated 1.0E+05 units/mL) and were extracted from 100 μL microorganisms' stocks using the Thermo Fisher viral RNA extraction kit (PureLink™ Viral RNA/DNA Mini Kit) or Qiagen QIAamp DNA Mini Kit. Extracted RNA/DNA were eluted to 100 μL with sterile RNase-free water. 5.5 μL of the purified RNA/DNA samples was used for each reaction and tested using the QuantiVirus™ SARS-CoV-2 Test Kit. The cross-reactivity testing results are summarized in S3 Table.

The cross-reactivity tests were run in triplicate, and all test controls passed (Positive controls Ct<25, No target control Ct >45, Extraction control has RP Ct <28). The tested organisms were all negative for the targeted N, E and ORF1ab genes of SARS-CoV-2 and human RP gene, indicating there is no cross-reactivity between SARS-CoV-2 primers & probes and any of the comparison organisms tested. The cross reactivity with common Human coronaviruses and MERS-coronavirus was also tested, and there was no cross reactivity at $10^5$ PFU/mL.

## Assay sensitivity verification

We spiked non-infectious viral particles (SeraCare AccuPlex SARS-CoV-2 Reference Material Kit) into healthy donor saliva which were confirmed by SARS-CoV-2 qPCR test to be negative, and tested each using three different qPCR instruments, ABI QuantStudio 5, ABI 7500 Fast Dx and BioRad CFX 384.

Clinical evaluation of the QuantiVirus™ SARS-CoV-2 Test was conducted with saliva specimens including 40 contrived positive and 30 negative samples. Saliva samples collected from healthy donors were mixed with the MGI RNA extraction kit's lysis buffer at 1:1 ratio before spiking in non-infectious viral particles for contrived positive sample (SeraCare AccuPlex SARS-CoV-2 Reference Material Kit) (S4a–S4c Table).

Table 2. Results of clinical NPS sample evaluation using QuantiVirus™ SARS-CoV-2 test.

| Patients samples | N | SARS-CoV-2 | | Sensitivity (95% CI) | Specificity (95% CI) | PPV (95% CI) | NPV (95% CI) |
|---|---|---|---|---|---|---|---|
| | | Detected | Not Detected | | | | |
| Positive | 85 | 84 | 1 | 98.8% (0.927–0.999) | 100% (0.949–1.00) | 100% (0.946–1.00) | 98.9% (0.931–0.999) |
| Negative | 90 | 0 | 90 | | | | |

* ABI QuantStudio 5 qPCR instrument was used for this test.

20 contrived positive saliva samples were created with the addition of non-infectious viral particles templates at 1x estimated LOD (1x200 copies/mL), 10 saliva samples were spiked at 1.5xLoD (300 copies/mL), and another 10 saliva samples were spiked at 2.5xLoD (500 copies/mL). Viral RNA was extracted from spiked samples and tested with the QuantiVirus™ SARS-CoV-2 test kit. The results showed that all 40 spiked saliva samples tested positive and all 30 control saliva samples tested negative on all three PCR instruments (S4a–S4c Table) with 100% (95% CI: 0.76–0.99) for positive call rate and 100% (95% CI: 0.91–1.00) for negative call rate.

## Clinical evaluation on NPS samples

Using the QuantiVirus™ SARS-CoV-2 Test, we tested clinical NPS samples including 85 positive samples and 90 negative samples which previously had been tested on Abbott m2000 molecular system using Abbott Real-Time SARS-CoV-2 testing kits (Table 2). The data shows that the clinical sensitivity of QuantiVirus™ SARS-CoV-2 test is 98.8% (95% CI: 92.7% -99.9%) and specificity is 100% (95% CI: 94.9%-100%). Its PPV is 100% (95% CI: 94.6%-100%) and NPV is 98.9% (95% CI: 93.1%-99.9%).

## Clinical evaluation of paired NPS and saliva sample

We tested and evaluated the concordance between paired NPS and saliva samples collected from patients by QuantiVirus™ SARS-CoV-2 tests. Among the 20 pairs of nasopharyngeal swabs (NPS) and saliva samples, the test results were the same for 16 pairs (16/20, 80% concordance rate), including 5 positive pairs and 11 negative pairs. There were four samples for which the results were discordant (S5 Table). Of these four pairs, one pair was NPS positive and saliva negative, whereas the other three pairs were NPS negative and saliva positive. Nevertheless, we compared NPS and saliva specimens by Wilcoxon matched pair signed rank test. The two samples types show 80% concordance with no significant differences (p = 0.13, Fig 2A), and its RP were similar between two types of specimens (p = 0.06, Fig 2B).

## Comparison of QuantiVirus™ SARS-CoV-2 multiplex kit with FDA EUA approved Abbott Realtime SARS-CoV-2 kit

We tested 24 saliva samples of recovering COVID-19 patients with the QuantiVirus™ SARS-CoV-2 kit in comparison with the Abbott m2000 RealTime SARS-CoV-2 PCR kit in parallel (Table 3). Data showed a concordance of the assays of about 88%. There were three samples detected by QuantiVirus™ SARS-CoV-2 kit, but not detectable with the Abbott kit (patients #8, 11 and 12), consistent with the reported higher sensitivity of QuantiVirus™ SARS-COV-2 PCR assay [22].

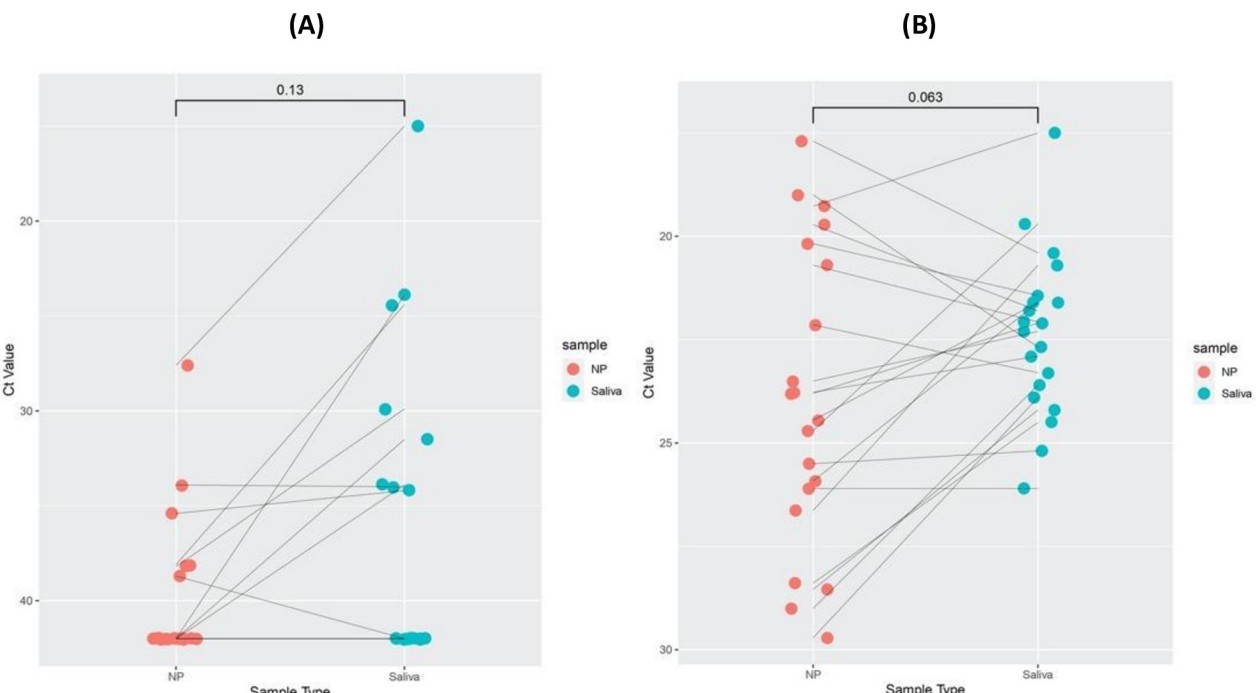

**Fig 2. Clinical evaluation of paired nasopharyngeal and saliva samples.** Cycle threshold (Ct) values for target gene and RP gene of NPS and saliva specimens were compared by Wilcoxon matched pairs signed rank test. Cycle threshold (Ct) values for viral E gene, N gene, O gene (ORF1ab), and human RNase P gene (RP) for NP and saliva specimens. A) E, N, and O Ct values for paired NP and saliva samples. Pairs are connected by a line. The Ct was set to 42 for samples in which signal was not detected. Ct values of E, N, and O were comparable between the two types of samples by Wilcoxon signed rank test. NPS and saliva concordance is about 80% with no significant differences ($p = 0.13$). B) RP Ct values for NP and saliva specimens were similar between the two types of samples by Wilcoxon signed rank test.

## Population screening using saliva samples

We tested 389 total saliva specimens collected from the general population of asymptomatic individuals (ie, asymptomatic screening) in Los Angeles and the San Francisco Bay Area counties. The screened population was represented by African Americans, White, Asian, and Latinx, with ages ranging from 18 to 80 (average 41) years old. From May 8 to Aug 26, 2020, 301 saliva samples were tested, and 5 samples were tested positive for SARS-CoV-2 by the QuantiVirus™ SARS-CoV-2 test. The 5 positives corresponded to 4 males of ages 19, 51, 52 and 54, and 1 female of age 34. Overall detection rate was 1.66% (Table 4). In another testing run of 88 saliva samples, 2 samples were positive and 86 were negative, with an overall positive detection rate of 2.27%. Together, we had screened 389 people from the general population and found that 7 people were positive for SARS-CoV-2 with an overall detection rate of 1.8%, consistent with the reported average positive testing rate from the same periods in the two metropolitan regions.

## Evaluation of pooling saliva samples for SARS-CoV-2 screening

To test the feasibility of pooling saliva specimens for screening asymptomatic patients, we pooled negative and positive saliva samples, and tested a total of 77 pooled positive samples (1 patient sample mixed with 5 healthy saliva samples; 1:6 ratio) and 54 pooled negative samples (mixed 6 healthy samples) (Table 5). Of the 77 pooled positive saliva samples, 73 were tested positive (average Ct of three genes: O gene Ct ~29.8; E gene 30.9 and N gene Ct ~31,0) and 4

**Table 3. Comparison of Abbott m2000 SARS-CoV-2 PCR test and DiaCarta QuantiVirus™ SARS-CoV-2 PCR test for SARS-CoV-2 detection in clinical saliva samples.**

| Method Comparison | Abbott m2000 Real-time SARS-CoV-2 | | Diacarta QuantiVirus SARS-CoV-2 multiplex | | | | |
|---|---|---|---|---|---|---|---|
| | Accession # | Detection & qPCR Ct | Detection | ROX(E-Gene) | Cy5(N-Gene) | FAM(ORF 1ab gene) | VIC(RP Gene) |
| Patient 1 | Saliva 1 | Not Detected | Not Detected | Undetermined | Undetermined | Undetermined | 21.6 |
| Patient 2 | Saliva 2 | Not Detected | Not Detected | Undetermined | undetermined | Undetermined | 22.9 |
| Patient 3 | Saliva 3 | Not Detected | Not Detected | Undetermined | undetermined | Undetermined | 22.1 |
| Patient 4 | Saliva 4 | Not Detected | Not Detected | Undetermined | undetermined | Undetermined | 21.6 |
| Patient 5 | Saliva 5 | Detected (Ct 18.21) | Detected | 32.1 | 31.7 | 30.0 | 20.7 |
| Patient 6 | Saliva 6 | Detected (Ct 31.00) | Detected | Undetermined | 37.8 | Undetermined | 23.7 |
| Patient 7 | Saliva 7 | Not Detected | Not Detected | Undetermined | Undetermined | Undetermined | 19.7 |
| Patient 8 | Saliva 8 | Not Detected | Detected | 37.4 | 37.3 | 42.4 | 23.3 |
| Patient 9 | Saliva 9 | Detected (Ct 23.89) | Detected | Undetermined | 24.4 | undetermined | 21.8 |
| Patient 10 | Saliva 10 | Not Detected | Not Detected | Undetermined | undetermined | Undetermined | 22.3 |
| Patient 11 | Saliva 11 | not detected | Detected | 32.8 | 33.9 | 31.5 | 23.9 |
| Patient 12 | Saliva 12 | not detected | Detected | 35.7 | 38.5 | 33.9 | 24.2 |
| Patient 13 | Saliva 13 | Not Detected | Not detected | Undetermined | 43.5 | Undetermined | 24.5 |
| Patient 14 | Saliva 14 | Detected (Ct 20.77) | Detected | 36.1 | 37.0 | 34.0 | 26.1 |
| Patient 15 | Saliva 15 | Detected | Detected | 35.6 | 35.1 | 34.2 | 23.6 |
| Patient 16 | Saliva 16 | Not Detected | Not detected | Undetermined | 37.7 | Undetermined | 23.6 |
| Patient 17 | Saliva 17 | Not Detected | Not detected | Undetermined | Undetermined | 41.2 | 32.7 |
| Patient 18 | Saliva 18 | Not Detected | Not detected | Undetermined | Undetermined | Undetermined | 24.9 |
| Patient 19 | Saliva 19 | Not Detected | Not detected | Undetermined | Undetermined | 33.5 | 29.1 |
| Patient 20 | Saliva 20 | Detected (Ct. 21.40) | Detected | Undetermined | 39.4 | 36.8 | 26.6 |
| Patient 21 | Saliva 21 | Not Detected | Not detected | Undetermined | Undetermined | Undetermined | 25.1 |
| Patient 22 | Saliva 22 | Not Detected | Not detected | Undetermined | 43.5 | Undetermined | 23.6 |
| Patient 23 | Saliva 23 | Not Detected | Not detected | Undetermined | Undetermined | Undetermined | 29.1 |
| Patient 24 | Saliva 24 | Not Detected | Not detected | Undetermined | Undetermined | Undetermined | 25.2 |

was reported as undetected. The average internal control (IC) RP Ct was 21.9 for all 131 pooled samples. Positive Predictive Value (PPV) is 100% (95% CI: 93.8%-100%). Negative Predictive Value (NPV) is 93.1% (95% CI: 82.5–97.8%). Additionally, we tested a total of 49 pooled positive saliva samples, created by mixing 1 patient sample with 11 healthy samples (1:12 ratio). Of the 49 pooled positive samples, 44 were tested positive (O gene, E gene and N gene average Ct 31.8, 32.1 and 31.9) and 5 was reported as undetected. Its IC RP average Ct was 22.3 for all 49 pooled saliva samples and additional 20 pooled healthy saliva samples. PPV is 100% (95% CI: 89.9%-100%) and NPV is 80.0% (95% CI: 58.7%-92.4%), respectively.

## Discussion

We have developed and validated a multiplex rRT-PCR assay for SARS-CoV-2 detection in saliva samples with clinical sensitivity of 98.8% (95% CI: 92.7%-99.9%) and specificity of 100%

**Table 4. Summary of saliva-based COVID-19 screening using QuantiVirus™ SARS-CoV-2 test in local communities.**

| Date | Total (N) | Positive | Negative | Detection Rate (%) |
|---|---|---|---|---|
| May 8-Aug. 26, 2020 | 301 | 5 | 296 | 1.66% |
| Aug. 28, 2020 | 88 | 2 | 86 | 2.27% |
| Total | 389 | 7 | 382 | 1.80% |

**Table 5. Saliva sample pooling for SARS-CoV-2 detection by QuantiVirus™ SARS-COV-2 test kit.**

| Saliva Sample Pooling | Sample Test (N) | Positive | Negative | Total Screen Sample(N) | Sensitivity | Specificity | PPV (%) | NPV (%) |
|---|---|---|---|---|---|---|---|---|
| 1 positive + 5 negative pooling | 77 | 73 | 4 | 462 | 94.8% (95%CI: 0.865–0.983) | 100% (95% CI: 0.917–1.00) | 100% (95% CI: 0.938–1.00) | 93.1% (95%CI: 0.825–0.978) |
| 6 negative pooling | 54 | 0 | 54 | 324 | | | | |
| 1 positive + 11 negative pooling | 49 | 44 | 5 | 588 | 89.8% (95% CI: 0.769–0.962) | 100% (95% CI:0.799–1.00) | 100% (95% CI: 0.899–1.00) | 80.0% (95% CI: 0.587–0.924) |
| 12 negative pooling | 20 | 0 | 20 | 240 | | | | |

*Single positive sample Mean Ct~30.3 at ORF1ab, Ct~22.3 RP; 1:6 ratio positive samples Mean Ct ~31.1 at ORF1ab, Ct 21.9 at RP; 1:12 ratio positive sample Mean Ct~32.3 at ORF1ab, Ct 23.3 at RP.

**Total screen sample (N) = pooling samples number x pooling ratio.

(95% CI: 94.9%-100%). Its PPV is 100% (95% CI: 94.6%-100%) and NPV is 98.9% (95% CI: 93.1%-99.9%). The detection of three viral target genes in one PCR tube enables a high throughput test using RT-qPCR. For these validated 384-well plate PCR platforms, 381 patient samples can be tested in each run (plus 3 controls). We have validated and integrated MGISP-960 high-throughput Automated Sample Preparation System, which can extract 192 samples (2x96) in about 80 min. For a CLIA laboratory with two MGI-960 machines, 380 samples can be tested with results available within 4 hrs. (Fig 1B).

We spiked SARS-CoV-2 viral particles into healthy donor saliva and confirmed that the analytical sensitivity (LOD) of the QuantiVirus™ RT-qPCR test is ~100 copies/mL for Bio-Rad CFX 384 and ~200 copies/mL for ABI QS5, ABI 7500Dx and Roche LC 480. Comparing to other FDA approved test kits, we have confirmed that our test kit has 600 NAAT Detectable Units/mL (NDU/mL) by FDA Reference Panel Testing and is among the top of all FDA approved SARS-CoV-2 test kits (S6 Table and FDA EUA website [19]). The multiplex RT-qPCR test can simultaneously detect three viral gene targets, which can minimize false negative results as chances of simultaneous mutations in all three target genes in the viral genome are highly unlikely. Furthermore, the results confirm that human saliva samples do not inhibit the RT-qPCR reaction, possibly due to the fact that inhibitor-tolerant RT-PCR master mix was used in the QuantiVirus™ SARS-CoV-2 test kit.

Leung et al. [6] analyzed 95 patient-matched paired samples from 62 patients including 29 confirmed patients with COVID–19 and 33 COVID–19 negative patients. The concordance rate was 78.9% (75/95 samples) between NP and saliva. Vogels et al. [20] reported a positive agreement of 83.8% (31/37 positive samples) for nasopharyngeal swabs and saliva when using TaqPath COVID-19 combo kit. Our data showed 80% concordance and no significant differences between NP and saliva, which is consistent with Leung's & Vogels's reports. Interestingly, for patient #4, the viral RNA was detected in the NPS sample by both Abbott m2000 test and DiaCarta test, but viral RNA was not detected in the saliva by either test. For patient #6, viral RNA was not detected in the NPS sample by either test, whereas viral RNA was detected in the saliva by both tests. This observation suggests that sample collection variabilities such as the time of collection and NPS sample quality do matter in SARS-CoV-2 PCR testing.

The 20 paired NPS-saliva samples were collected from recovering patients being evaluated prior to their release from self-quarantine. Consequently, their viral loads were much lower (100–1000 copies/mL), compared to the viral loads expected for the initial diagnostic testing. For patients with active SARS-CoV-2 infection (viral loads typically are above 10,000 copies/mL), there should be no problem detecting the virus in adequately collected saliva samples.

Landry et al. [21] described that most of saliva samples from sick patients were thick, stringy, and difficult to pipet. Since we used the QuantiVirus™ SARS-CoV-2 saliva sample collection kit which has VTM solution in the collection tube, the saliva was diluted 2-fold and therefore much easier to process. Matic et al. [22] used PBS at a 1:2 dilution that also helped resolve highly viscous saliva samples; however, manual dilution after collection may be associated with pipetting errors and cross contamination.

The QuantiVirus™ SARS-CoV-2 test results were 87.5% in concordance with FDA EUA approved Abbott RealTime SARS-CoV-2 results for saliva samples, with a higher detection rate overall. In fact, this observation is consistent with recently reported test sensitivity among various SARS-CoV-2 molecular tests. FDA published its SARS-CoV-2 Reference Panel Comparative Data on its website on Sept 15, 2020 [19]. It reported that QuantiVirus™ SARS-CoV-2 Kit has LOD of 600 NDU/mL whereas Abbott Realtime SARS-CoV-2 assay has LOD of 2700 NDU/mL. Accordingly, the reason for the observation that SARS-CoV-2 viral RNA was detected in three patient samples by the QuantiVirus™ SARS-CoV-2 test but not by Abbott RealTime SARS-COV-2 assay was likely due to the higher sensitivity of the QuantiVirus™ SARS-CoV-2 assay. It also demonstrated that saliva specimens represent a viable specimen type that can be easily applied for COVID-19 testing when using more sensitive tests.

A total of 389 saliva specimens from the general population were tested and demonstrated the feasibility of using saliva for large scale population screening. Saliva is a non-invasive and easily collectable specimen for COVID-19 screening. Given the drawbacks of nasopharyngeal and oropharyngeal swab sample collection, saliva sampling could be applied as an acceptable alternative [23].

With saliva pooling strategy, we have demonstrated that 6-samples pooling (1 patient mixed with 5 healthy saliva samples, or 1:6 ratio) has 94.8% sensitivity (95% CI: 86.5–98.3%) and 100% specificity (95% CI:91.7–100%), As noted, of the 77 pooled saliva samples, 4 pooling samples were tested negative. In fact, for these 4 pooled samples, the individual positive samples used for the pooling had Ct of 34.4, 34.8, 35.7 and 37.5 for ORF1ab gene, respectively, consistent with low viral loads to start with (less than 100–200 copies/mL) (see Table 1a–1d). Therefore, in order to detect weakly positive patient in pooled samples, a RT-PCR test with LOD at 100–200 copies/mL or higher is required. If pooling testing is considered, each clinical laboratory should establish laboratory-specific pooling protocol based on the LOD of SARS-CoV-2 molecular test. One advantage of pooling testing is its cost-effectiveness, allowing population-based asymptomatic screening or monitoring even when testing supplies are limited.

In summary, we have demonstrated that saliva specimens can be reliably used for SARS-CoV-2 detection, and saliva-based large-scale population screening for COVID-19 with or without pooling is feasible.

## Supporting information

**S1 Table. Tentative LOD determination by series dilution.**
(DOCX)

**S2 Table.** a. Intra assay precision of the QuantiVirus SARS-Cov-2 test kit. b. Operator reproducibility of the QuantiVirus SARS Cov-2 test kit. c. Inter-instrument precision of the QuantiVirus SARS CoV-2 test kit.
(DOCX)

**S3 Table. Results of cross-reactivity evaluation of the QuantiVirus™ SARS CoV-2 test kit.**
(DOCX)

**S4 Table.** a. Evaluation of contrived saliva samples with added non-infectious viral particles (Bio-Rad CFX 384). b. Evaluation of contrived saliva samples with added non-infectious viral particles (ABI QuantStudio 5). c. Evaluation of contrived saliva samples with added non-infectious viral particles (ABI 7500 Fast Dx).
(DOCX)

**S5 Table. Paired NPS and saliva samples tested by QuantiVirus™ SARS-CoV-2 test.**
(DOCX)

**S6 Table. Assay sensitivity evaluation of various FDA EUA approved SARS-CoV-2 RT-PCR test kits.**
(DOCX)

# Acknowledgments

The authors thank Dr. Ramanathan Vairavan and Eric Chu for reviewing and editing this manuscript, and Christopher Teixeira, Fady Ettnas, Enas Eltarazy and Eric Abbott for their technical contributions.

# Author Contributions

**Conceptualization:** Chuanyi M. Lu, Michael Y. Sha.

**Data curation:** Jonathan Li, Hui Ren, Larry Pastor, Yulia Loginova, Kristopher Taylor.

**Formal analysis:** Zhao Zhang.

**Investigation:** Jonathan Li, Hui Ren, Larry Pastor, Yulia Loginova, Joseph K. Wong, Aiguo Zhang, Michael Y. Sha.

**Methodology:** Qing Sun, Michael Y. Sha.

**Project administration:** Aiguo Zhang.

**Software:** Zhao Zhang.

**Supervision:** Qing Sun, Roberta Madej, Joseph K. Wong, Aiguo Zhang, Chuanyi M. Lu, Michael Y. Sha.

**Validation:** Roberta Madej, Michael Y. Sha.

**Writing – original draft:** Qing Sun, Chuanyi M. Lu, Michael Y. Sha.

**Writing – review & editing:** Joseph K. Wong, Aiguo Zhang, Chuanyi M. Lu, Michael Y. Sha.

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
