## [Decision Letter · Decision Letter 0]

18 Dec 2020

PONE-D-20-36825

Saliva as a testing specimen with or without pooling for SARS-CoV-2 detection by multiplex RT-PCR test

PLOS ONE

Dear Dr. Sha,

Thank you for submitting your manuscript to PLOS ONE. After careful consideration, we feel that it has merit but does not fully meet PLOS ONE’s publication criteria as it currently stands. Therefore, we invite you to submit a revised version of the manuscript that addresses the points raised during the review process.

We look forward to receiving your revised manuscript.

Kind regards,

Ruslan Kalendar, PhD

Academic Editor

PLOS ONE

Journal Requirements:

"This study not involve any funded"

"no any competing interests"

We note that one or more of the authors are employed by a commercial company: 1DiaCarta Inc. 2600 Hilltop Dr. Richmond, CA 94806.

(2) Please also provide an updated Competing Interests Statement declaring this commercial affiliation along with any other relevant declarations relating to employment, consultancy, patents, products in development, or marketed products, etc.  

4. We note that Table 3A, 3B, and 3C have been previously published in https://www.fda.gov/media/136809/download. Thus, we require specific consent from the copyright holder to publish these tables in PLOS ONE, under the CC BY 4.0 license. To seek permission from the copyright owner to publish these tables under the Creative Commons Attribution License (CCAL), CC BY 4.0, please contact them with the following text and PLOS ONE Request for Permission form (http://journals.plos.org/plosone/s/file?id=7c09/content-permission-form.pdf):

“I request permission for the open-access journal PLOS ONE to publish XXX under the Creative Commons Attribution License (CCAL) CC BY 4.0 (http://creativecommons.org/licenses/by/4.0/). Please be aware that this license allows unrestricted use and distribution, even commercially, by third parties. Please reply and provide explicit written permission to publish XXX under a CC BY license.”

Please upload the granted permission to the manuscript as a Supporting Information file. In the caption of the copyrighted tables, please include the following text: “Republished from [ref] under a CC BY license, with permission from [name of publisher], original copyright [original copyright year].

Reviewers' comments:

Reviewer's Responses to Questions

**Comments to the Author**

1. Is the manuscript technically sound, and do the data support the conclusions?

Reviewer #1: Partly

Reviewer #2: Yes

Reviewer #3: Yes

Reviewer #4: Yes

2. Has the statistical analysis been performed appropriately and rigorously? 

Reviewer #1: No

Reviewer #2: Yes

Reviewer #3: Yes

Reviewer #4: Yes

3. Have the authors made all data underlying the findings in their manuscript fully available?

Reviewer #1: Yes

Reviewer #2: Yes

Reviewer #3: Yes

Reviewer #4: Yes

4. Is the manuscript presented in an intelligible fashion and written in standard English?

Reviewer #1: Yes

Reviewer #2: Yes

Reviewer #3: Yes

Reviewer #4: Yes

5. Review Comments to the Author

Reviewer #1: The manuscript entitled “Saliva as a testing specimen with or without pooling for SARS-CoV-2 detection by multiplex RT-PCR test.” by Sun and colleagues described their evaluation of a FDA-EUA commercial assay for SARS-CoV-2 detection by multiplex RT-PCR using saliva as a specimen type. Also, several papers have already been published on detection of SARS-CoV-2 in saliva, this particular kit was not evaluated for saliva specimens. The manuscript is generally well written, but it has several major concerns with data presentation and conclusions made:

Major concerns:

In the method section, the authors described how the have designed the primers and probes for the assay although their sequences were not provided. Also, it appears Taqpath mastermix was used for RT-qPCR. It is not clear if these are part of the components/methods for the commercial kit they have described?

Figure 2: Tt is not clear what is the relevance of the ROC curve? ROC is used to determine cut-off or threshold that gives best sensitivity and specificity. How the ROC curve and AUC was calculated? The current figure suggests decreasing specificity with increasing sensitivity! Also the AUC does not seem be not correct!

Table 1 and 3: Data analyzed and presented in such a manner which is inappropriate for evaluation of performance characteristics of a clinical assay. The limit of detection in saliva should be empirically determined with proper statistics. The authors may refer to the guidelines in this review article: https://www.ncbi.nlm.nih.gov/pmc/articles/PMC2901657/

Also presenting data in Table 3 as PPA and NPA, as part of clinical evaluation, is not correct because these are all spiked samples with known quantity of the analyte, within the analytical range, they are all expected to be detected anyway. However, these results may be presented to determine inhibition/interference from saliva specimens (if different saliva samples were utilized), and may also be used to determine intra-assay and inter-assay variability using PCR CT values.

Table 2: Why RP is negative in most samples? How many samples for each target was tested? How SD and CI were calculated?

Table 6: unnecessary repeat in different columns - detected vs not detected

CT significantly weaker in the positive samples by Quantivirus compared to the reference method but there were 3 extra positive results by Quantivirus needs further verification to ensure that they were not false positive results. Also, it is not clear how the results of prospective evaluation with only 24 samples in this case compares with their retrospective data in Table 4 in terms of sensitivity and specificity?

Table 8 is confusing. Poos better expressed as the pool of 1 pos + 5 neg specimen. How the specimens were tested. How does the CT values compare between single vs pooled runs? Because pooled runs were made using pre-selected samples, results are better to express as % agreement instead of sensitivity/specificity/PPV/NPV

Minor concerns:

Page 13 last paragraph: pooled health sample – check grammar in all the places this term was used.

Too many tables with detailed raw data. Show data summary instead and add supplemental tables if necessary.

Reviewer #2: 

The authors demonstrate that saliva specimens can be reliably used for SARSCoV-2 detection, and saliva-based large-scale population screening for COVID-19 with or without pooling is feasible. I think this is important to help many people who have problems, to do nose picking and get answers easily and safely, overcoming the drawbacks of nasopharyngeal (NPS) or oropharyngeal swabs (OPS) swabs. According to the statistics produced in this paper, the clinical performance of saliva-based tests is comparable to that of NPS-based clinical tests.

Reviewer #3: 

The manuscript describes the validation of both non-pooled and pooled saliva samples compared to the gold std of NP samples. The data is very good and supports the conclusions made. This is an important manuscript given the relative ease of sample collection, the amenability to high throughput and pooling and the accuracy of the results and should be published.

i had only one question re the diluting of the saliva samples: i thought the denominator (2nd #) was supposed to indicate the # of total dilutions rather than the # of diluent (healthy) samples as in titers for serology. Thus, the first ratio should be 1:6 and the 2nd a 1:12 - ? a minor point.

Reviewer #4: 

This is a nice study to extend the use of a validated SARS-Cov-2 multiplex RT-PCR assay for detection in saliva. Overall, it is well written, thorough, and the appropriate statistical analyses have been included.

MAJOR Concerns:

1. Financial Disclosure: Clearly this study was supported (people, reagents, instruments, etc...) and funded by DiaCarta. This should be included on the financial disclosure.

2. Competing Interests: Nine out of the twelve authors are associated, i.e. employed, by DiaCarta, LLC. who is the manufacturer/developer of this QuantiVirus kit. As stated in the Plos ONE Competing Interests policy, employment is considered a financial competing interest and authors should provide details on the relationship to the funder (i.e. employment) and a "Description of funder’s role in the study design; collection, analysis, and interpretation of data; writing of the paper; and/or decision to submit for publication."

The fact that these items were not included, and even submitted with the manuscript with statements of... "This study not involve any funded" and "no any competing interests" is the biggest concern I have regarding publication.

MINOR Concerns and Comments:

General:

1. Please double check to ensure all acronyms are spelled out the first time they are used. MGI, CLC, UNG, ROC, etc..

Introduction:

1. Some description of the saliva collection devices and process, in general as well as from other studies would be useful. Is the collection from stimulated saliva or passive drool? Are comparable volumes collected?

2. Reference is needed near the bottom of page 4 following... "Other researchers also reported that SARS-CoV-2 was detected in 91.7% (n=11) of the initial saliva specimens from confirmed COVID-19 patients. (REF)"

Methods:

1. All of the methods generally lack detail, particularly with respect to the sample pooling and analytical sensitivity. Typically the analytical sensitivity involves a serial dilution of a known concentration or spiking of negative samples with a known amount of viral RNA at decreasing concentrations. Then the LOD is established as the 0 copies/mL + 3 standard deviations. If this or another method was used it needs to be described clearly and with sufficient detail that someone/anyone could repeat it and obtain the same result.

Results/Discussion:

1. How does the analytical sensitivity of 100-200 copies/mL compare to other test kits (multi-plex and non)? It may be helpful to include an example graph of the analytical sensitivity data rather than just a summary table.

2. On page 12, last sentence in section titled "Clinical evaluation of paired NPS and saliva samples" you should remove the plural (s) from "samples types"

3. On page 13, in the section titled "Population screening using saliva samples" the dates of collection do not correspond to what is written in the Methods section...was this population screening included in the IRB exemption? The way it is written sounds like the samples were collected specifically for this study and not excess/otherwise discarded. Additional clarification may be useful.

4. On page 13, last paragraph third line from the bottom "health" should be "healthy" in two instances...this also occurs in Table 8 left hand column.

5. Additional clarity is needed to describe the pooled samples (back in the Methods section would be best) because it is unclear if 77 distinct pooled samples from unique donors (1 patient + 5 healthy individuals) were created and used. Also, it is not clear why there was a different number of 1+5 pooled samples vs. 1+11 pooled samples were analyzed (N=77 vs N=49).

6. Table 8, what is the column labeled "Sample Screen(N)" there doesn't seem to be any mention or reference to that anywhere else in the manuscript, but maybe I missed it.

7. First line of your Discussion...I would add "We have developed and validated a multiplex RT-qPCR assay for SARS-CoV-2 detection in saliva..." so that you can emphasize the distinction of this work from previous assay validation in NPS and its significance.

8. Page 15, third line from the top...I would consider re-wording "On the other hand" phrase as it sounds negative and the statement is a quite positive one regarding the lack of inhibition from saliva.

6. PLOS authors have the option to publish the peer review history of their article (what does this mean?). If published, this will include your full peer review and any attached files.

Reviewer #1: No

Reviewer #2: No

Reviewer #3: **Yes: **Melissa Kennedy

Reviewer #4: No

---

## [Author Response · Author response to Decision Letter 0]

28 Jan 2021

5. Review Comments to the Author

Reviewer #1: The manuscript entitled “Saliva as a testing specimen with or without pooling for SARS-CoV-2 detection by multiplex RT-PCR test.” by Sun and colleagues described their evaluation of a FDA-EUA commercial assay for SARS-CoV-2 detection by multiplex RT-PCR using saliva as a specimen type. Also, several papers have already been published on detection of SARS-CoV-2 in saliva, this particular kit was not evaluated for saliva specimens. The manuscript is generally well written, but it has several major concerns with data presentation and conclusions made:

Major concerns:

In the method section, the authors described how the have designed the primers and probes for the assay although their sequences were not provided. Also, it appears Taqpath mastermix was used for RT-qPCR. It is not clear if these are part of the components/methods for the commercial kit they have described?

Yes, it is part of the commercial kit

Figure 2: Tt is not clear what is the relevance of the ROC curve? ROC is used to determine cut-off or threshold that gives best sensitivity and specificity. How the ROC curve and AUC was calculated? The current figure suggests decreasing specificity with increasing sensitivity! Also the AUC does not seem be not correct!

An ROC space is defined by false positive rate (FPR) and true positive rate (TPR) as x and y axes, respectively. Since TPR is equivalent to sensitivity and FPR is equal to 1 − specificity, the ROC graph is sometimes called the sensitivity vs (1 − specificity) plot. The best possible prediction method would yield a point in the upper left corner or coordinate (0,1) of the ROC space, representing 100% sensitivity (no false negatives) and 100% specificity (no false positives). Since our curve indicate “specificity” but not “1-specificity”, it is why it sets from 1.0 to 0.0 in x-axis. we plotted the ROC curve and calculated the AUC with R package “pROC”. When using normalized units, the area under the curve (often referred to as simply the AUC) is equal to the probability. AUC value is high, then its probability is high. 

In order to avoid any confusion, we decide to delete this Figure (Fig 2).

Table 1 and 3: Data analyzed and presented in such a manner which is inappropriate for evaluation of performance characteristics of a clinical assay. The limit of detection in saliva should be empirically determined with proper statistics. The authors may refer to the guidelines in this review article: https://www.ncbi.nlm.nih.gov/pmc/articles/PMC2901657/

It was our negligence does not descript clear in the manuscript. In fact, we used empirical method to determine our LOD. Below is how we decide our assay analytical sensitivity (LOD): 

To determine the Limit of Detection (LoD) and analytical sensitivity of the QuantiVirus SARS CoV-2 Test kit, studies were performed using serial dilutions of analyte and the LoD was determined to be the lowest concentration of template that could reliably be detected with 95% of all tested positive. LoD of each target assay in the QuantiVirus™ SARS-CoV-2 Test were conducted and verified using SeraCare AccuPlex SARS-CoV-2 Reference Material Kit (Cat# 0505-0126). Non- infectious viral particles from the AccuPlex SARS-CoV-2 Reference Material Kit were spiked in saliva at various concentrations (50 copies/mL, 100 copies/mL and 200 copies/mL) diluted from the stock concentration of 5000 copies/mL. Real- time RT-PCR assay was performed with the provided kit reagents and tested triplicate on ABI QS5, ABI 7500 Fast Dx, Bio-Rad CFX 384 PCR and Roche LightCycler 480 II instruments. Then the LOD was confirmed by testing 1xLoD of viral RNA with 20 replicates. The LoD was determined to be the lowest concentration (copies/ml) at which ≥95% (19/20) of the 20 replicates were tested as positive. We summary these data as supplementary table 1 and clarify in page 11. 

Also presenting data in Table 3 as PPA and NPA, as part of clinical evaluation, is not correct because these are all spiked samples with known quantity of the analyte, within the analytical range, they are all expected to be detected anyway. However, these results may be presented to determine inhibition/interference from saliva specimens (if different saliva samples were utilized), and may also be used to determine intra-assay and inter-assay variability using PCR CT values.

Respond: agree. We delete this PPA and NPA calculation. We have updated our intra-assay and inter-assay data for this assay precision, page 11

Table 2: Why RP is negative in most samples? How many samples for each target was tested? How SD and CI were calculated?

Because these samples are pure viral RNA or bacteria genome DNA. That is why their RP is negative except our internal extraction control and positive control gblock. 

Table 6: unnecessary repeat in different columns - detected vs not detected

CT significantly weaker in the positive samples by Quantivirus compared to the reference method but there were 3 extra positive results by Quantivirus needs further verification to ensure that they were not false positive results. Also, it is not clear how the results of prospective evaluation with only 24 samples in this case compares with their retrospective data in Table 4 in terms of sensitivity and specificity?

This is a good point. The samples were tested freshly in UCSF hospital by reference method (Abbott qPCR kit) and tested in our CLIA lab one month late by couple of thaw and freeze. That may explain why the Ct is higher by our kit. The extra 3 positive samples by QuantiVirus kit were repeatedly test and confirmed positive. We discuss this on our discussion section. One reason is Abbott kit has lower LOD (2700 NDU/mL) compare to our kit (600 NDU/mL) See detail in our new S6 Table. Assay sensitivity evaluation of various FDA EUA approved SARS-CoV-2 RT-PCR test Kits. And discuss on page 18

Table 8 is confusing. Poos better expressed as the pool of 1 pos + 5 neg specimen. How the specimens were tested. How does the CT values compare between single vs pooled runs? Because pooled runs were made using pre-selected samples, results are better to express as % agreement instead of sensitivity/specificity/PPV/NPV

We have updated the method section for how to be pooling these samples; Single sample average Ct 30.3 at ORF1ab; 1: 6 pooling Ct 31.3 and 1: 12 pooling Ct ~32.3 although RP Ct 22.3 to 21.9 and 23.3. Since we try to know how the assay sensitivity during sample pooling, a sensitivity and specificity maybe the better expression.

Minor concerns:

Page 13 last paragraph: pooled health sample – check grammar in all the places this term was used.

thanks, we correct it

Too many tables with detailed raw data. Show data summary instead and add supplemental tables if necessary.

Agree, we have moved some tables to supplementary section.

Reviewer #2: 

The authors demonstrate that saliva specimens can be reliably used for SARSCoV-2 detection, and saliva-based large-scale population screening for COVID-19 with or without pooling is feasible. I think this is important to help many people who have problems, to do nose picking and get answers easily and safely, overcoming the drawbacks of nasopharyngeal (NPS) or oropharyngeal swabs (OPS) swabs. According to the statistics produced in this paper, the clinical performance of saliva-based tests is comparable to that of NPS-based clinical tests.

Thanks for the comments.

Reviewer #3: 

The manuscript describes the validation of both non-pooled and pooled saliva samples compared to the gold std of NP samples. The data is very good and supports the conclusions made. This is an important manuscript given the relative ease of sample collection, the amenability to high throughput and pooling and the accuracy of the results and should be published.

i had only one question re the diluting of the saliva samples: i thought the denominator (2nd #) was supposed to indicate the # of total dilutions rather than the # of diluent (healthy) samples as in titers for serology. Thus, the first ratio should be 1:6 and the 2nd a 1:12 - ? a minor point.

Yes, agree. We should use 1:6 and 1:12 dilution. We have corrected it on our manuscript. 

Reviewer #4: 

This is a nice study to extend the use of a validated SARS-Cov-2 multiplex RT-PCR assay for detection in saliva. Overall, it is well written, thorough, and the appropriate statistical analyses have been included.

MAJOR Concerns:

1. Financial Disclosure: Clearly this study was supported (people, reagents, instruments, etc...) and funded by DiaCarta. This should be included on the financial disclosure.

We have updated this in our cover letter.

2. Competing Interests: Nine out of the twelve authors are associated, i.e. employed, by DiaCarta, LLC. who is the manufacturer/developer of this QuantiVirus kit. As stated in the Plos ONE Competing Interests policy, employment is considered a financial competing interest and authors should provide details on the relationship to the funder (i.e. employment) and a "Description of funder’s role in the study design; collection, analysis, and interpretation of data; writing of the paper; and/or decision to submit for publication."

The fact that these items were not included, and even submitted with the manuscript with statements of... "This study not involve any funded" and "no any competing interests" is the biggest concern I have regarding publication.

We have updated our cover letter with “There is no conflict of interest associated with this publication, and there has been no financial support for this study that could have influenced its outcome. The authors received no funding for this work. Diacarta Inc provided salary support for authors QS, HR, JL, LP, RM, YL, ZZ, AZ and MS. The specific roles of these authors are articulated in the ‘author contributions’ section. This commercial affiliation (to Diacarta Inc) does not alter our adherence to PLOS ONE policies on sharing data and materials”.

MINOR Concerns and Comments:

General:

1. Please double check to ensure all acronyms are spelled out the first time they are used. MGI, CLC, UNG, ROC, etc..

Thanks, we correct it.

Introduction:

1. Some description of the saliva collection devices and process, in general as well as from other studies would be useful. Is the collection from stimulated saliva or passive drool? Are comparable volumes collected?

Stimulated saliva was collected from patient with collection kit (see left picture of figure 1). 

Yes, we collected about 2 mL saliva in collection tube with 2 mL VTM buffer and used 200 uL as viral RNA extraction. We update this with briefly statement in the method of our manuscript.

2. Reference is needed near the bottom of page 4 following... "Other researchers also reported that SARS-CoV-2 was detected in 91.7% (n=11) of the initial saliva specimens from confirmed COVID-19 patients. (REF)"

We have updated it in manuscript “other researchers 13 also……”

Methods:

1. All of the methods generally lack detail, particularly with respect to the sample pooling and analytical sensitivity. Typically the analytical sensitivity involves a serial dilution of a known concentration or spiking of negative samples with a known amount of viral RNA at decreasing concentrations. Then the LOD is established as the 0 copies/mL + 3 standard deviations. If this or another method was used it needs to be described clearly and with sufficient detail that someone/anyone could repeat it and obtain the same result.

This is good point. We have updated a method how to establish the LOD in this assay (see manuscript page 9)

Results/Discussion:

1. How does the analytical sensitivity of 100-200 copies/mL compare to other test kits (multi-plex and non)? It may be helpful to include an example graph of the analytical sensitivity data rather than just a summary table.

This is a good question. In fact, our kit was confirmed as 600 NDU/mL and is top 4 product of all FDA approved kits when all kits were tested by FDA Reference Panel Blind Testing. Comparing to Thermo Fisher multiplex kit (its LOD 180000 NDU/mL), this kit has better sensitivity. We have a Supplementary Table 5 for these kits’ comparison.

2. On page 12, last sentence in section titled "Clinical evaluation of paired NPS and saliva samples" you should remove the plural (s) from "samples types"

Good catch, thanks

3. On page 13, in the section titled "Population screening using saliva samples" the dates of collection do not correspond to what is written in the Methods section...was this population screening included in the IRB exemption? The way it is written sounds like the samples were collected specifically for this study and not excess/otherwise discarded. Additional clarification may be useful.

Thanks for point it out. It is typing error; it should be May-Sept 2020. We have corrected it on page 6.

4. On page 13, last paragraph third line from the bottom "health" should be "healthy" in two instances...this also occurs in Table 8 left hand column.

Correct, thanks

5. Additional clarity is needed to describe the pooled samples (back in the Methods section would be best) because it is unclear if 77 distinct pooled samples from unique donors (1 patient + 5 healthy individuals) were created and used. Also, it is not clear why there was a different number of 1+5 pooled samples vs. 1+11 pooled samples were analyzed (N=77 vs N=49).

We have edited in the method for this pooling.

We total applied 77 individual patient samples pooling with 385 healthy samples by 1:6 ratio mixed to create 77 pooling positive samples. And we applied 324 healthy samples by 1:6 ration to pooing into 54 pooling negative samples.

The reason for 1:6 (N=77) and 1:12 (N=49) is what we want to balance the target screening sample # between 400-600.

6. Table 8, what is the column labeled "Sample Screen(N)" there doesn't seem to be any mention or reference to that anywhere else in the manuscript, but maybe I missed it.

Thanks reminder. We have added a sentence in Method page 7 and indicate how to calculate this number under table 8 (new table 5) 

7. First line of your Discussion...I would add "We have developed and validated a multiplex RT-qPCR assay for SARS-CoV-2 detection in saliva..." so that you can emphasize the distinction of this work from previous assay validation in NPS and its significance.

Good suggestion. Thanks. We have accepted this emphasize sentence. 

8. Page 15, third line from the top...I would consider re-wording "On the other hand" phrase as it sounds negative and the statement is a quite positive one regarding the lack of inhibition from saliva.

Thanks, we charge to “furthermore”.

---

## [Editor Report · Decision Letter 1]

1 Feb 2021

Saliva as a testing specimen with or without pooling for SARS-CoV-2 detection by multiplex RT-PCR test

PONE-D-20-36825R1

Dear Dr. Sha,

We’re pleased to inform you that your manuscript has been judged scientifically suitable for publication and will be formally accepted for publication once it meets all outstanding technical requirements.

Kind regards,

Ruslan Kalendar, PhD

Academic Editor

PLOS ONE

---

## [Editor Report · Acceptance letter]

11 Feb 2021

PONE-D-20-36825R1 

Saliva as a testing specimen with or without pooling for SARS-CoV-2 detection by multiplex RT-PCR test 

Dear Dr. Sha:

I'm pleased to inform you that your manuscript has been deemed suitable for publication in PLOS ONE. Congratulations! Your manuscript is now with our production department. 

Kind regards, 

on behalf of

Prof. Ruslan Kalendar 

Academic Editor

PLOS ONE